# Health care providers' perceptions and experiences related to Midwife-led continuity of care–A qualitative study

Solomon Hailemeskel[1,2]*, Kassahun Alemu[3], Kyllike Christensson[4], Esubalew Tesfahun[5], Helena Lindgren[4]

1 School of Midwifery, College of Medicine and Health Sciences, University of Gondar, Gondar, Ethiopia, 2 Department of Midwifery, College of Health Science, Debre Berhan University, Debre Berhan, Ethiopia, 3 Department of Epidemiology and Biostatistics, Institute of Public Health, College of Medicine and Health Sciences, University of Gondar, Gondar, Ethiopia, 4 Division of Reproductive Health, Department of Women's and Children's Health, Karolinska Institute, Solna, Sweden, 5 Department of Public Health, College of Health Science, Debre Berhan University, Debre Berhan, Ethiopia

* solomonhailemeskel9@gmail.com

**Data Availability Statement:** All relevant data are within the manuscript and its Supporting Information files.

## Abstract

### Background

Though Midwife-led care remains a key to improving the health status of pregnant mothers, in Ethiopia, maternity care has traditionally been based on a model in which responsibility for care is shared by hospital-based midwives, nurses, general practitioners, and obstetricians. This type of care has been seen as representing a fragmented approach.

### Objective

The aim of this study was to explore health care providers' perceptions and experiences related to Midwife-led continuity of care at primary hospitals in the north Shoa zone Ethiopia

### Methods

A qualitative approach was selected as the methodology for this study. Data were collected from 25 midwives and 8 integrated emergency surgical officers (IESO) and medical doctors working in maternal health care units in four primary hospitals in the north Shoa zone, Amhara Regional State. Four focus group discussions and eight individual interviews were conducted. The facilitator utilized a set of open-ended questions for the focus group discussion. Semi-structured interview questions were used for the interviews and thematic data analysis was done.

### Finding

The main theme extracted was "Midwives welcome consideration of a Midwife-led model that would provide greater continuity of care, but they expressed concerns about organisation and workload". The midwives said that they would welcome working with the midwife-led care model, as they believed using it could lead to improving the quality of maternal

**Funding:** The author(s) received no specific funding for this work.

**Competing interests:** The authors have declared that no competing interests exist.

health care, provide greater continuity, and improve coverage, birth outcomes, and maternal satisfaction. The midwives could become more autonomous and be able to take more responsibility for maternity care. The group of 25 midwives and the group of 8 IESO and medical doctors perceived that working procedures and changes in the organization of care in the health facility would have to be studied carefully before any changes can be considered.

## Conclusion

In this study, we found that replacing the existing system of maternal care with a Midwife-led model would require careful analysis of how this model of care might be implemented in Ethiopia. Further investigation will be of great importance in providing insights that will help in developing a final model.

## Introduction

In many countries, midwives are the primary providers of maternity care [1] following what is known as the Midwife-led model of care. In other countries, an obstetrician has primary responsibility for care following the shared model [2]. These are two basic models of maternal care, each practiced with variations [2, 3]. In those countries where the Midwife-led model is the norm, continuity of care is provided by a trusted and known midwife or by a small group of midwives (midwifery team) that supports a woman, during the antenatal, intrapartum and postnatal periods to facilitate a healthy pregnancy and childbirth, and healthy parenting practices [2, 4].

Midwife-led continuity of care (MLCC) is a well-researched health systems innovation that has been shown to lead to better outcomes, such as lower rates of prematurity and foetal/neonatal death [2], fewer medical interventions [5], better experiences among women and lower costs compared to other models of maternal and new-born care [6, 7]. MLCC models consist of three elements: 1) Care is provided by midwives and they are the lead carers throughout pregnancy, birth and the postnatal period; 2) Continuity of care by a known midwife who works in collaboration with other health professionals 3) Care is based on the midwifery philosophy that pregnancy and birth are physiological life events and is therefore focused primarily on preventive and supportive care rather than exclusively on the identification and treatment of risks and complications [2]. The model also includes identifying, referring and coordinating care for women who require obstetric or other specialist attention. Thus, the MLCC model exists within a multidisciplinary network in which consultation and referral to other care providers occur when necessary [8].

Over the past decades, the standard model of care has contributed greatly to improving outcomes for both mother and child [2, 9]. In shared models of care responsibility for the organisation and delivery of care, throughout initial booking to the postnatal period, is shared between different health professionals depending on the stage of pregnancy [2, 3] Studies of European maternity care following the standard model have however found that unnecessary interventions are still practiced in low-risk pregnancies [2, 10]. Decisions to avoid giving what some see as unnecessary interventions remain a subject for discussion [10]. To help deal with these issues, alternative models of care have been developed as noted above, models that promote acceptance of the view that pregnancy and childbirth are normal parts of life [10, 11].

Health care providers and policy makers in many countries have been reviewing the existing policies on maternal health care in an effort to determine if the midwife-led model of care might be employed as an alternative to past practice [2, 10, 12].

In Ethiopia, modern maternity care has been organized based on a model in which responsibility for care is shared by hospital-based midwives, nurses, general practitioners, and obstetricians. This type of care has been seen by some as a fragmented approach and has even been referred to as an impersonal "conveyor belt" style of care [2, 3, 9].

The organization of maternity care in Ethiopia has long been determined by the number of midwives and nurses available in a health facility. In most Ethiopian health facilities, antenatal care has long been provided by the primary-care health service where nurses are in charge [9]. Ideally, if it could become possible to provide enough midwives to make it possible for all antenatal care, labour, delivery and postnatal care to be given by midwives it might be possible to consider adoption of the MLCC model [2, 13]. If this could be done, then a Midwife-led model could be analysed in order for a trial to be carried out [2]. If the trial was found to be successful, the model could be more widely adopted.

Postnatal follow-up is divided into two intervals: During the first two hours after delivery, postnatal care is given by midwives in a postnatal ward. After the first two hours, care is given by a clinical nurse and/or midwives at a postnatal clinic. If there are not enough midwives available to provide all the care needed in the labour and birth wards, nurses can be assigned to conduct labour and delivery care as well. Health officers, general practitioners, emergency surgical officers and obstetricians are to be available to aid as needed. When this happens, a mother may encounter exposure to several different health professionals. In the midwife-led model, care would be based on relational continuity in which the mother would be in contact with the same person over time [9, 13, 14].

Midwife-led continuity of care is now considered as the key to improving the health status of pregnant mothers [2, 15]. In Ethiopia, the staff of maternal and child health care units have different backgrounds and experience as health care providers. Any consideration of modifying the existing system must be preceded by exploring the views of all who are involved, having them describe their experience and their perceptions about midwife-led care. The aim of this study was, therefore, to explore health care providers' (midwives, IESO and medical doctors) perceptions and experiences related to Midwife-led continuity of care at primary hospitals in the north Shoa zone Ethiopia

## Methods

### Study design and period

A descriptive qualitative study design was used to explore health care providers' perceptions and experiences related to Midwife-led continuity of care. We used this approach in an effort to learn what the participants in the two groups of maternity health care providers say about this model and its possible applications in Ethiopia [16, 17]. The study was conducted from September 7, 2019 to October 25, 2019.

### Study area

This study was conducted in four primary hospitals in the North Shoa Zone, Amhara Regional State. Currently, more than two million people reside in the zone [18]. In the zone, there is one referral hospital linked with 8 primary hospitals, 95 health centers and 389 health posts. There are 10 to 15 midwives in each primary hospital and 24 midwives in Debre Berhan referral hospital. Each primary hospital has a minimum of two integrated emergency surgical officers but an obstetrician and/or gynaecologist are assigned in the referral hospital only. There are also

practicing medical doctors and health officers in each primary hospital. The primary hospitals are located at the "woreda" level (the lowest administrative unit in Ethiopia) and are expected to provide service for a population of 100,000 and to provide a capacity of 100 beds. The hospitals are expected to provide all health services. According to the 2017/18 report of the zone's health departments there were 75,039 reported pregnancies. Of these, 62,974 (84%) mothers to be had only the first ANC visit and 43,565 (58%) had all four ANC visits. Of that total, 39,665 (53%) of the women gave birth at one of the health facilities and the remainder gave birth at home [18].

## Study participants

All health care providers who were in service at the time of the beginning of the study and who had worked in maternal and child health units for the preceding 6 months and were willing to participate (32 midwives, 10 medical doctors and 8 emergency surgical officers) were considered as eligible for the study and were thus invited to participate. A total of 25 midwives, 4 medical doctors and 4 emergency surgical officers consented to participation.

## Sample size

Generally, for qualitative studies there is no pre-determined method for determining the 'required' sample size. One group of researchers suggested between 3 and 10 participants [19, 20], another up to 20 [21] and yet another between 3 and 30 participants [22]. Clearly, there is no agreement on what is best. In any case, there must be a sufficient number of participants to be able to identify the full range of views and understandings [23]. By considering the point of saturation, we used a total of 33 participants, 25 midwives and 8 IESO and medical doctors. The data collection was completed when no new/unique information was emerged.

## Data collection

Four focus group discussions were held with the midwives and eight in-depth interviews were held with members of the non-midwife group. All the data collection was conducted face-to-face at the health facilities of the study participants. The principal investigator (SH) and two trained data collectors who do have MSc in maternal and reproductive health with previous experience on qualitative data collection conducted the focus group discussions with the midwives and also carried out the interviews with the group-2 members IESO and medical doctors. After reviewing relevant literature, an interview guide for individual interviews and discussion guide for the focus group discussion were developed and pretested prior to carrying out the first interview. A one day training was given for the data collectors on how to interview, record and transcribe the data. The focus group discussions lasted for one hour, and the interviews lasted for an average time of 15 minutes and took the form of a guided conversation. The discussion was initiated by asking the study participants to explain about the type of model of maternal health care practiced in their hospitals and how do they understand model of maternal health care? Then the interview continued based on the participants response by using probing questions until we gate reach data on the research question we want to answer. In each study area, before the interview started, the importance of confidentiality among the discussants was addressed. All the study participants were given both written and oral information about the objectives of the study and confidentiality of information was assured. Both the principal investigator and the data collectors do not have any relationship with the study participants. All the study participants gave their oral informed consent. The discussions were conducted in Amharic, the local language in the area and all the discussions and interviews were audio recorded with the consent of the participants.

## Data analysis

The process of data analysis was inspired by the thematic analysis method as described by Braun and Clarke [24]. Phase 1) Interviews (data items) were transcribed verbatim by the principal investigator (SH) together with the data collectors who conducted the interview, then read and re-read several times to get familiarized with the data set. The translated data were cross-checked with the audio file to ensure its proper transcription and translation. In the stage of transcription and reading, some patterns in the data were identified. Phase 2) Meaningful features across the data set were coded and data related to each code were compared. A list with different codes (325 initial codes) was developed from the entire data set based on the study aim. Code and meaning units were identified by three of the investigators (SH, HL and KC) and we set together and decide on the identified meaning units. Phase 3) In this stage, codes were sorted into potential broader subthemes/themes and related data were gathered within each possible subtheme/theme. Phase 4) All coded data extracts were re-read and their structure under each subtheme/theme was reviewed. Those forming a consistent pattern within subthemes/ themes were examined. A thematic map of analysis was generated. Phase 5) Data extracts that formed subthemes and themes were re-examined, organised, and refined. The essence of each subtheme/theme was identified and explained to readers in five subthemes. The subthemes were grouped together according to their content and these subthemes generated two themes. The themes and subthemes that emerged were supported by participant quotations, which were italicized in the text with a unique identification code indicating the participant code and working hospitals.

## Ethical approval

Ethical approval was obtained from the Institutional Review Board of University of Gondar (ref no: O/V/P/RCS/05/1050/2019). Permission letters were obtained from the regional health bureau, the zonal health department and the hospital administration. An informed and signed consent was obtained from each participant. Participants were informed that their participation in this study was voluntary, the information they gave would remain confidential and would be used only for research purposes. Participants could withdraw from participating in the study at any time.

# Findings

In all four-study hospitals, 25 midwives and 8 medical directors and IESO participated. Participants were between 23 and 33 years of age with a mean of 26.76 ± 2.79 years with level of education diploma, degree and master's degree. The study participants had an average of 4.12± 2.31 years of working experiences and 7 of them were females (Table 1).

## Number and type of themes identified

The findings consist of one overarching theme: *"Midwives welcoming continuity of care despite concerns about organisation and workload" "As a professional midwife I would be happy if I can give all antenatal, labour and delivery and postnatal care for mothers using the continuum of care model. But, the reality is if I am assigned at antenatal care clinic I would only provide antenatal care; the other unit of care will be covered by other professionals" (midwife code 1 Mehal Meda Hospital). And* two themes and five sub themes (Fig 1).

Theme A: "Fragmented organization of midwifery care" with its subthemes describes the organization of midwifery care at each "home" hospital, how the midwives work in the maternal and child health care units, and how they organize themselves at all different levels. The

**Table 1. Sociodemographic characteristics of participants at primary hospitals, north Shoa, Amhara Region, Ethiopia, 2019 (n = 33).**

| Characteristics | Number (percent) |
|---|---|
| **Gender** | |
| Female | 7 (21%) |
| Male | 26 (79%) |
| **Age (mean±SD) 26.76 ± 2.795** | |
| 20–25 | 14 (42.42%) |
| 26–30 | 16 (48.48%) |
| 31–35 | 3 (9.1%) |
| **Profession** | |
| Midwife | 25 (75.76%) |
| Medical Doctor | 4 (12.12%) |
| Integrated Emergency Surgical Officer (IESO) | 4 (12.12%) |
| **Years of experience in maternal health care (mean±SD) 4.12 ±2.315** | |
| < 3 years | 7 (21.21%) |
| 3–5 years | 15 (45.46%) |
| ≥ 5 years | 11 (33.33%) |
| **Level of education** | |
| Diploma Midwife | 4 (12.12%) |
| Degree Midwife | 21 (63.64%) |
| Medical Doctor | 4 (12.12%) |
| Masters /specialization (IESO) | 4 (12.12%) |
| **Working unit for midwives** | |
| ANC | 5 (20%) |
| Delivery | 18 (72%) |
| PNC | 1 (4%) |
| Family planning | 1 (4%) |
| **Working unit for Medical doctors and IESO** | |
| All maternal and child health unit | 8 (100%) |

emergent subthemes indicate understanding of the model of maternal health care and organization of the midwives in the health facilities.

Theme B: "Perceived role of midwife-led continuity of care in the Health Care System" with its subthemes describes the role and contribution of Midwives in the health care system. The emergent subthemes identified were midwife-led continuity of care and quality of care, contribution of Midwife-led continuity of care for the health of the mothers and midwife-led continuity of care and development of Midwifery profession.

**A. Fragmented organization of midwifery care.** This theme describes the organization of midwifery care at all levels. The midwives described the model of care practiced in Ethiopia as fragmented; the midwife-led model of care has not been used. Different health care providers are responsible for the care of the mother from pregnancy up to postnatal period. Most of the time, midwives are not assigned to participate or be present at each unit of care because of the shortage of midwives and policy problems. The midwives explained that in order for a midwife-led model of care to be established and employed, it would first be necessary to establish a policy supported by an implementation manual. In addition, the existing working environment and infrastructure would all have to be improved.

*A.1. Understanding on model of maternal health care.* The participants perceived that the phrase "model of maternal health care" refers to any one of a number of systems of care for

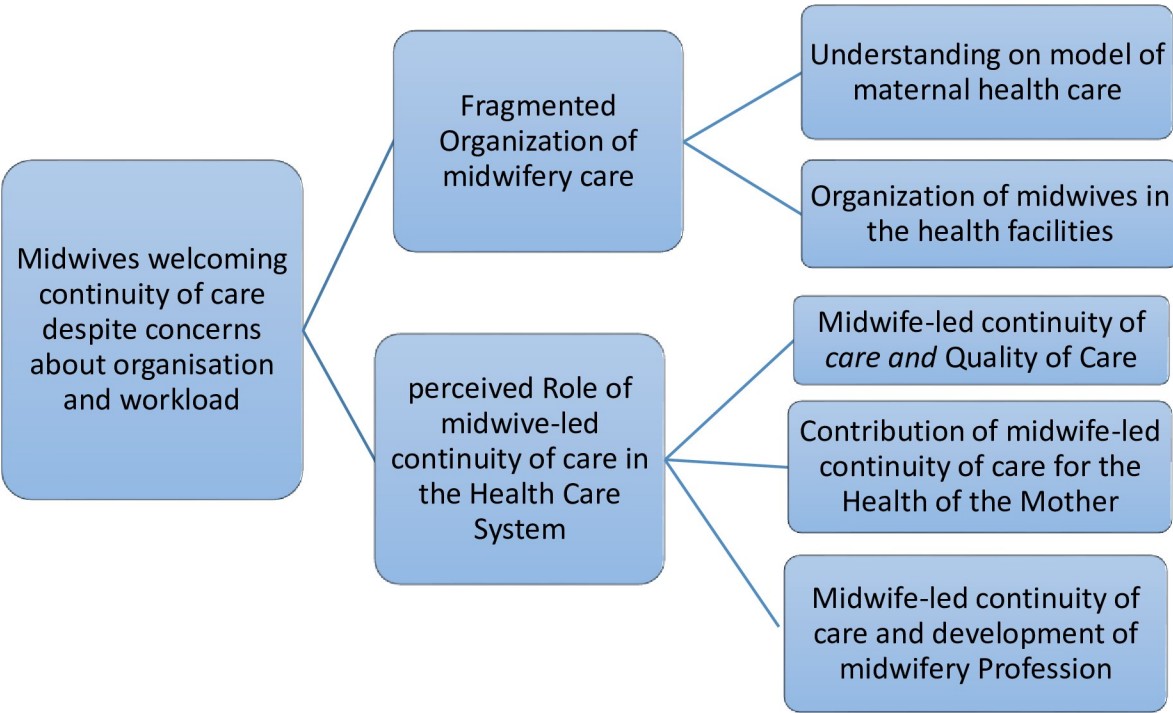

**Fig 1. Main theme, themes and subthemes identified in the analysis process.**

pregnant mothers starting from their entry into antenatal care and continuing into postnatal period. They were familiar with different systems and referred to them as: 1) The focused Antenatal care model (ANC), 2) the client centred care model, and 3) the midwifery model (where midwives cover the whole maternal health care service).

The participants showed conflicting views when they were asked about the continuity of care model and what continuity of care means. Some participants said that continuity of care means that midwives are available in all units of maternal health but that the mother might meet different midwives. Others thought that continuity of care provider means that mother meets the same midwife at first ANC visit and followed all the way through PNC.

The participants stressed the importance of considering the specific goals that must be met to establish a midwife-led continuity of care model. Of these goals, development of higher-level policy and guidelines that can support this model is the most important. They pointed out that for the successful implementation and utilization of this model, supportive policy and clear guidelines are needed.

The midwives paid special attention to how setting the midwife-to client ratio (workload) would be determined. The number of midwives per health facilities should be optimum to implement the Midwife-led model:

" *When I was at ANC, I could meet all mothers. However, I cannot get them at delivery unless they came when I was on duty. By any means, if she finds me at the delivery room, she insists and asks me to attend her delivery. However, I cannot do it because the system does not allow me to do it as a different Midwife assigned for this purpose*" (midwife code 1 and 5, Mehal Meda Hospital).

*A.2. Organization of midwives in the health facilities.* The midwives explained how they work in a 6-months or 3-months rotation between ANC, labour and delivery, postnatal care

and family planning and how this might hamper any effort to employ the midwife-led approach.

> "*In our hospital, we midwives are assigned from ANC up to PNC and work in a rotation (shift) in every six month period; we are not following the same mothers from ANC up to PNC by the same individual, rather the mother rotates across the units*" (midwife code 4 Alemketema and code 6 Shoa Robit Hospital).

In other words, different midwives are involved at each stage as the mother passes from one stage to the next. Due to shortage of midwives on duty, midwives are assigned in the labour and delivery rooms only. In all types of existing care in Ethiopia, it is rare for the mother to be in contact with the same midwife at every stage.

The midwives understood that if the Midwife-led system were to be tested or adopted they would have to take on major responsibility for planning and organization the system. *"For me, model of maternal health care is considered when the mother received all ANC, delivery and postnatal care in continuous manner in the same health facility by same professional." (Integrated emergency surgical officer Ataye Hospital).*

**B. Perceived role of midwife-led continuity of care in the health care system.** This theme describes the role of midwives in the health-care system. The midwives said that the midwife-led care model could improve the quality of maternal health care; the midwife would be more autonomous and practice at all levels and midwifery professional development as well. Successful implementation of the model needs effective planning, communication with concerned bodies, and collaboration and teamwork on the part of the professionals. Leaders should have a clear implementation strategy, staff and organizational ownership, and professional autonomy for those working in the model. The implementation strategy of the model should be tailored to the local context with the necessary human and financial resources.

*B.1. Midwife-led continuity of care and quality of care*. The midwives said that having midwife-led continuity care might improve the quality of maternal and child health care in Ethiopia. Participants believe that the quality of maternal care could be improved if they are given responsibility from start to finish. The philosophy of midwife-led care should be women centred and encourage normal birth through reducing unnecessary interventions. As a result, midwife-mother bonding, maternal satisfaction, and the continuity of care will be improved.

This quotation illustrates this: *"In continuity of care model, I will give equal emphasis for each unit of care because I am responsible for each unit of care" (midwife code 2 Alemketema Hospital).*

However, the participants said that in the existing model of care, care is fragmented. The system causes difficulties for the mother since she does not know who will be responsible for her at any point in the entire process. This may lead her to express dissatisfaction with the process.

> "*In shared model of care the mothers even did not know who provided the care, when we asked them, they told us by his/ her colour (the one chocolate face, the black guy)." (Medical doctor Shoa Robit Hospital).*

The relationship created between the mother and the provider is limited and only unit based not covering the whole process of care, which results in loss of information.

> "*As the mother moves does not meet the same health care provider in every service she required during pregnancy, the quality of care supposed a mother needs getting compromised.*

*There will be loss of information because the mother clinical profile might not be documented."* (Midwife code 5 Ataye Hospital).

The unit-based arrangement of midwives had also another problem.

*". . .The ANC service does not contribute for the improvement of institutional delivery. For example, ANC coverage nationally is 65% but only 27% of them had history of institutional delivery. Implies, little or no attention* may be given *for birth and post-natal care."* (Integrated emergency surgical officer Alemketema hospital).

**B.2. Contributions of Midwife-led care for the health of a mother.** The participants highlighted that the positive contributions of midwife-led care for the health of the mother result from the equal emphasis given for each unit of care:

*"If the service is organized in a continuity of care model, I will give equal emphasis for each unit of care because I am responsible for each unit of care. In addition, with this model of care I will have my own pregnant mothers in which I am supposed to follow up to PNC.As a result of this the service uptake will be improved and unnecessary interventions will be reduced."* (Midwife code 2 Alemketema Hospital).

The participants believed that the new model would also help the mother to become cooperative in aiding in care management, and that the birth process and outcome of delivery would be improved.

*"It will be great for the women to always have the same midwife in her pregnancy life and it will be very good for her to know the midwife before having the baby. Because knowing the midwife that looked after her during the childbirth process will help the women to feel safe and trust on the providers* (midwife code 2 Ataye hospital).

This was also well explained by the IESO and medical doctors. They pointed out that they will be happy to have a model of care that improves the service and outcomes of care provided by midwives. *"Midwife-led continuity of care mode is good. In this type of model of care, the labouring woman would be more relaxed and cooperative for her care. If the woman has no complication beyond the midwives' competency it is better not to have any intervention by other professionals."* (Integrated emergency surgical officerShoa Robit Hospital).

When the mother meets the midwife on a regular basis, she becomes more at ease and feels free to discuss issues with the provider and voluntarily do what the midwife tells her to do.

*"I am working in ANC and repeated contact with mothers, the mothers used to call me by my name and ask me for any information freely. We used to discuss as brotherly and sisterly and consult me about their health. I believe this result in better maternal satisfaction."* (Midwife code 1 Shoa Robit Hospital).

The participants pointed out that mothers were happy, relaxed and cooperative during labour.

*"I was in the delivery ward and* one client came *to it and she got me and she was relaxed, happy and smile when she* knew *that I was the one who* was *responsible* for *attending her delivery. She* was *happy and relaxed because she* knew *me during her ANC visit, I* provided *all four ANC care for her."* (Midwife code 1 Ataye Hospital).

The same feeling was reported by the IESO and medical doctors. This provider pointed out that even if it is a must to transfer the women to obstetricians or emergency surgical officers, the women will come with clear indication which makes her care management easy."*When transfer of the women to obstetrician or medical doctors was needed, the women from the midwifery care were much easier to work with because the woman had learnt a lot about her pregnancy and birth from her continuity of care provider (midwife)."(Medical doctor Alemketema Hospital)*. They attributed this to the advantages of continuity of care model in which the model helps the woman to better understand the reason for her transfer.

*B.3. Midwife-led continuity of care and development of midwifery profession.* The participants believe that midwife-led care will help the midwife to work more autonomously and independently. *"I will be happy if we have a model of care that will allow us to work more autonomously, a model of care that allows midwives to work on all competencies" (midwife code 3 Ataye Hospital)*.

They believe that this model will lead to improvements in midwives' professional contribution toward the entire birth process. This model of care helps to improve midwives' confidence and allows them to manage all maternal health care independently and responsibly. The other points highlighted by the participants were that this kind of model of maternal health care helps the midwife to feel more responsible for the health of the mother, do all things in the best possible way, make care provision easy and successful, know everything about the mother and improve the provider's interest in providing care.

> *I would like to work with midwife-led continuity of care in the future as it is the ideal way I would like to practice and the care I want to be able to provide, as it gives me more confidence and autonomy in my work (midwife code 4 Alemketema Hospital).*

To practice in this model of care, the participants pointed out the importance of short term training to improve midwives' skill and knowledge of this model, making sure that this model can reduce the work burden of midwives be more effective. Besides, they expressed their concern about the implementation of Midwife-led continuity of care as the number of midwives may not be sufficient to keep the caseload at the appropriate level. They fear it may increase workload for midwives, and midwives may become fatigued and suffer from burnout which might affect team spirit.

## Discussion

The findings of this qualitative study provide a contemporary view of health care providers working in maternal and child health unit experiences of their practice and perception on the midwife-led continuity of care model in North Shoa Zone, Amhara Regional State, Ethiopia. The participants provided rich descriptions of their practice and view of the model of maternal health care. The participant midwives welcome the chance to work with midwife-led continuity of care, they expressed the view that the existed model (shared model of care) limits midwives' professional role and responsibilities. It causes information loss, fragmentation of care and stress for mothers. Besides, they pointed out importance of continuity of care for the health of the mother.

In this study, the participants welcomed the chance to work with the midwife-led continuity of care model as it gives them professional autonomy, develops midwives' self-confidence and improves the midwifery profession. This finding is found to be similar with results from previous studies which explained midwife-led continuity of models where the midwife is the lead professional and follows women starting from the initial booking appointment, up to and

including the early days of parenting or postnatal period [3, 8, 25, 26]. Participant midwives also mentioned that in midwife-led continuity of care the lead professional is the midwife in the planning and organizing the care of the women. This is similar to results from other studies in which the midwife is the pioneer in the provision of care for women with low risk [2, 3, 12, 27, 28]. In the previous studies and the present study continuity of care model is viewed as an excellent model for midwives as it allows them to work across their scope of practice and the model helps midwives to practise autonomously and by fully exploring their roles they will experience greater job satisfaction [29].

This study noted that in non-midwife-led continuity of care model, the role and responsibility of midwifery care is shared among different health care providers. Midwives do provide a significant amount of care but they are not autonomous in their practice. There are limitations on what they may do as has also been documented in prior studies were midwives account only 10% of the global sexual reproductive maternal new-born and adolescent health workforces [30, 31]. Similar findings were reported on the role of midwives as being diminished to the position of a doctor' assistant without any possibility to make individual clinical decisions and with very limited responsibility [2, 8]. These findings are quite different from the International Confederation of Midwives' stand that midwifery should be recognised as an autonomous profession in its scope of practice [32] and the World Health Organization recommendation to have midwife-led continuity of care [33]. The possible explanation for this challenge would be that the organizational model of care practiced in different countries has not been developed with reference to the philosophy of continuity of care model.

In the present study, consistent with other studies [3, 4, 34], midwives identified some of the disadvantages of the non-midwife continuity of care model; care becomes fragmented and unit based, non-continuity of care provider and information which could cause loss of information, because of the presence of new provider at each unit of care the mother feels stranger for the health care system and causes discomfort for her, getting the new provider at birth causes the mother to develop stress and affects the birth outcomes. On the other hand, the participants highlighted organizational policy; guideline and appropriate number of midwives should be there to work with midwife-led continuity of care model of care. In line with this, the participants highlighted their fear that midwife-led continuity of care may increase the workload of midwives, develops fatigue, burnout and affect team spirit. Similar to this study, other international studies had also discussed issues associated with midwife-led care model with midwives working on it and includes, burnout includes mixed day and night shifts [35, 36], working in isolation, high workload, long working hour and work life balance are among the influencing factors [35, 37–40]. Conversely, a recent Australian study found that midwives working in continuity of care models may benefit from midwife-led care model, with increased professional satisfaction and lower burnout scores when compared to their non-midwife-led care model colleagues [40].

The midwives participating in this study, pointed out that having midwife-led continuity of care improves maternal outcome, reduces unnecessary intervention, increase maternity service up-take and quality of maternal and child health care. A similar finding was reported in the Cochrane review on continuity of care through pregnancy, labour and birth by teams of midwives found that there were benefits for women who had continuity of midwifery care on positive birth experience [41, 42]. Women who received continuity of care by a known midwife had reported feeling better prepared for, more in control during labour [41, 42] and more pleased with their antenatal, intrapartum and postnatal care than women who received standard care [41]. For the health of the mother they highlighted that continiuty of care model will improve mother to midwife bonding and create friendly and trustworthy relationship between them. This was found to be similar with results from a study done previously as the central

aspects of midwifery care are related with helping women experience labour and birth in ways that can optimize her emotional well-being and promote a positive mother-infant interaction [43]. The goal is to make the woman feel confident, feeling cared for as a unique human being, and not considered as just another woman in the crowd are identified as important factors for establishing a trusting relationship between midwives and pregnant mothers [44].

A novel finding of this study is the different understanding the participants expressed regarding midwife-led continuity of care model in relation to continuity of care provider. Some thought that to say there is midwife-led continuity of care, there has to be a formal assignment of midwives in all-maternal and child health care unit. This group of midwives gave much emphasis to the availability of midwives in the ANC, labour and delivery care and postnatal care. Others feel that to say there is midwife-led continuity of care there has to be assignment and availability of the same midwife from ANC up to PNC and the mother should be able to get the same provider in all her visits. For this group of midwives, the most important component of continuity of care was the presence of the same provider (midwife) in all MCH units. The later concept of midwives is in line with the other studies' definition of the concept of midwife-led continuity of care. It is the presence of continuity of care giver, which refers to the presence of the same caregivers (midwife), that plans and provides most of the maternity care throughout pregnancy, labour, birth, and the postnatal period [2–4, 12].

## Strengths and limitations of the study

One limitation of the study is that no generalizations to larger populations can be made. A strength of the study was that all the interviews and focus group discussions were carried out in the participants' mother tongue-. The fact that the interviewers and the focus group discussants shared almost the same cultural and linguistic background eased the flow of the interviews and discussion and helped the participants to feel comfortable in sharing their experiences openly. The depth of our rich description of the results of the study from the experiences of the participants should enable readers to appraise the transferability of findings to varied model of maternal health care. One additional strength is that it was possible for the the research team to reach consensus about each theme and subtheme. Verbatim quotations perhaps will help readers to consider if they might share some of the opinions of the participants in our study.

## Conclusion and recommendation

This study is the first to describe the experience and perceptions of Midwives, IESO and medical doctors about the possible use of a different model of maternal and child health care than the traditional one that has been used in Ethiopia. The participants expressed their positive perceptions of the possibility of working with the Midwife-led continuity model of care and they believe that employing this model could have positive effects on the quality of maternal health care and would allow midwives to employ the full scope of their professional training and experience. They felt that well-established policy and guidelines would be needed and that it would be necessary to increase pay level for midwives commensurate with the increase in responsibility. They agreed that without an increase in staff numbers, the midwives' workload would increase. Further implementation research is needed to evaluate the effects for women, families and health care providers of implementing the midwifery-led continuity model of care in the Ethiopian context.

## Supporting information

**S1 Text. Focus group discussion and in-depth interview guide for health care providers.** (DOCX)

**S2 Text. Focus group discussion and in-depth interview participant's characteristics.**
(DOCX)

## Acknowledgments

We would like to thank the Midwives and other health care providers who participated in this study, as well as the participating health facilities for providing us place and access to the study participants.

## Author Contributions

**Conceptualization:** Solomon Hailemeskel, Kassahun Alemu, Kyllike Christensson, Esubalew Tesfahun, Helena Lindgren.

**Data curation:** Solomon Hailemeskel.

**Formal analysis:** Solomon Hailemeskel, Helena Lindgren.

**Investigation:** Solomon Hailemeskel.

**Methodology:** Solomon Hailemeskel, Kassahun Alemu, Kyllike Christensson, Esubalew Tesfahun, Helena Lindgren.

**Project administration:** Solomon Hailemeskel.

**Resources:** Solomon Hailemeskel, Esubalew Tesfahun, Helena Lindgren.

**Software:** Solomon Hailemeskel.

**Supervision:** Solomon Hailemeskel, Kassahun Alemu, Kyllike Christensson, Esubalew Tesfahun, Helena Lindgren.

**Validation:** Solomon Hailemeskel, Kassahun Alemu, Kyllike Christensson, Esubalew Tesfahun, Helena Lindgren.

**Visualization:** Kassahun Alemu, Kyllike Christensson, Esubalew Tesfahun, Helena Lindgren.

**Writing – original draft:** Solomon Hailemeskel.

**Writing – review & editing:** Solomon Hailemeskel, Kassahun Alemu, Kyllike Christensson, Esubalew Tesfahun, Helena Lindgren.

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
