## [Decision Letter · Decision Letter 0]

8 Mar 2021

PONE-D-20-34013

Health care providers’ perceptions and experiences related to Midwife-led continuity of care - a qualitative study

PLOS ONE

Dear Dr. Beshah,

Thank you for submitting your manuscript to PLOS ONE. After careful consideration, we feel that it has merit but does not fully meet PLOS ONE’s publication criteria as it currently stands. Therefore, we invite you to submit a revised version of the manuscript that addresses the points raised during the review process.

We look forward to receiving your revised manuscript.

Kind regards,

Bernadette Watson, Ph.D.

Academic Editor

PLOS ONE

Journal Requirements:

2. Please provide the following information in your Methods section relating to your qualitative methodology:

i) the expertise and training of the interviewers and

ii) how the focus groups and discussion groups audio recordings were translated for analysis.

Finally, please provide the interview guide used as a part of the study as Supporting Information.

3. Please remove your figure from within your manuscript file, leaving only the individual TIFF/EPS image file, uploaded separately.  This will be automatically included in the reviewers’ PDF.

Additional Editor Comments:

I have now received the comments from the two reviewers. They are both in agreement that your paper needs to be edited for conciseness. One of the reviewers raises concerns about confidentiality which you should address. I would urge you to take account of all their comments in your revision of the paper. Both reviewers note the importance of this area so I hope you will take on what is a large but worthwhile task.

I look forward to receiving your revisions.

Reviewers' comments:

Reviewer's Responses to Questions

**Comments to the Author**

1. Is the manuscript technically sound, and do the data support the conclusions?

Reviewer #1: Partly

Reviewer #2: Partly

2. Has the statistical analysis been performed appropriately and rigorously? 

Reviewer #1: N/A

Reviewer #2: N/A

3. Have the authors made all data underlying the findings in their manuscript fully available?

Reviewer #1: No

Reviewer #2: Yes

4. Is the manuscript presented in an intelligible fashion and written in standard English?

Reviewer #1: No

Reviewer #2: Yes

5. Review Comments to the Author

Reviewer #1: An interesting paper and important area of research. The manuscript is sound however it needs attention to the details of the study and justification of themes. Also need to be sure to include study details around questions and clarity in how these were developed and validated prior to the study.

Referencing needs to be consistent through the manuscript and quote original sources eg ref 1 quotes WHO guidelines for ANC to justify midwives as primary care providers – which is true but not from the guidelines (they include the primary reference in the guidelines). Need to reference all assertions.

Also consider the scope of your work as exploratory and provides further unknown information to inform delivery of maternity services (too much detail is provided in this paper about what that could be).

I have reviewed and noted the following points you may wish to consider to further improve your manuscript.

Introduction

Would be best to cite original research not guidelines ie midwives primary care providers in many countries – true so find original sources, and several to make this point. See earlier point

Explain what mid led care is – so it is clear to the reader.

Need references for all your assertions eg line 71 -74 need references for each of these you state

Line 80. Add what happens if antenatal deviates from usual physiological parameters (and reference)

Line 145 explain when you stop and how know enough for your sample (and reference). It is included but as last sentence, meaning justification and clarity could be improved with rewriting.

151 what were the themes of the questions; include example of the questions you asked. How did you determine these questions and test prior to the interviews?

These will help explain and describe your findings later – the link is not clear

163 do you mean written informed consent?

166 Data analysis process include but make briefer and no need for bolding of steps

186 include the approval number in the manuscript

198 explain what a primary hospital is – size, services – I may have missed this earlier?

201 Overarching theme is very broad (and includes sub section re workload).

Themes need justification as to their identification. Then having identify two themes – are these theme A & B. Be consistent with how describe ie keep as theme one (?A) and theme two (?B). You are very aware of the work it is new to readers so be sure to step through this clearly and methodically.

214 label the diagram as per your themes

247 need to edit and number these so themes and subthemes are very clear

285 + Code2 – is midwifery? Obstetric? Consider identifying the occupation for each

Reference list – include doi or online access information

Reviewer #2: I would like to thank the authors for the opportunity to review this manuscript. This is a very important topic with clear implications for practice, and I believe this work will be a valuable contribution to the literature. There are however, several issues with the manuscript itself that I believe need to be addressed before I can recommend it for publication.

General comments:

1. It is my opinion that the overall length of the manuscript (across all sections) could be reduced. Many sentences throughout could be abridged/shortened while retaining the same level of detail. Please consider this is your resubmission.

Introduction:

2. Page 4, row 74: please just use the acronym “MLCC” here as full name is already spelled out in row 71.

3. Page 4, row 81: please clarify what you mean by “the standard model of care”.

4. The authors discuss the ideal of making MLCC possible for all women. Therefore, I think a statement of how pregnancies deemed higher-risk (thus requiring the care of an obstetrician) would fit in to this model, is important.

5. As definitions of MLCC often differ, can the author’s please clarify their views on the number of midwives that would operate together/that each woman gets to know, in the model they are proposing? This is particularly important when considering what would happen if a woman’s known midwife was not available when needed– i.e., does the woman actually get to know a small group of midwives in the event her main midwife is not available for birth? If the author’s believe that these details should be determined at the point of MLCC implementation, please provide a statement clarifying this and/or discuss the different options that exist.

Methods:

6. Page 7, row 150: It is unclear if saturation was used to determine the sample size or if the authors just happened to reach saturation with the sample they were able to obtain. Please reword this sentence to clarify.

7. Page 7, row 156: please clarify what ‘Group-2’ is.

8. Data Collection: A brief description of the content of the interview and discussion guides is needed. It is difficult to interpret findings without knowing the types of questions that were asked/discussion points raised. e.g., did you provide the participants with a definition of MLCC? The full interview and discussion guides should also be provided as Appendices/Supplementary material.

9. Data Analysis: Please specify how many people were involved in each phase of the thematic analysis and whether duplicate coding was done etc.

Results:

10. I am worried that the results in current format may jeopardies participant confidentiality. For instance, the author’s note that all eligible emergency surgical officer’s (4/4) participated, and then each of the four participating hospitals are named in the results. For reader’s familiar with these hospitals, would it be obvious who these ESO’s were? There are also very low numbers (<5) of some participant groups outlined in Table 1. Please reconsider/revise some of the data presented in light of these potential confidentiality implications.

11. The authors refer often to ‘participants’ when discussing focus group/interview results. Here, it would be helpful to know specifically, if they are referring to midwives, doctors, or ESOs (or a combination of these). A strength of this work is that different types of providers were included. Ideally, this should enable the author’s to determine the concordance/discordance of views between these different provider groups.

12. The wording of the overarching theme ‘Midwives welcoming continuity of care despite concerns about organisation and workload’ comes across as being purely based on the midwives’ views and not inclusive of those in the other provider groups interviewed. I assume the overarching theme is based on the entirety of data analyzed, so please consider rewording this theme or explain if I have misunderstood something.

Discussion:

13. Page 18, row 375: There appears to be some repetition in this paragraph. Please review and revise.

14. Page 19, row 397: some errors in this sentence, please review and revise.

15. Page 21, row 449: “no generalizations to larger populations can be made” - please explain this further. Why do you think your sample was not representative of the general population of midwives, doctors, and ESOs in Ethiopia?

16. It would be interesting to know the author's views on the effectiveness of the focus groups vs. interviews? Do you think they yielded the same quality of data? Based on this, do you have recommendations for optimal methodology of future work?

17. More specific recommendations for next steps (e.g., What questions still need to be answered? What type of studies should be conducted?) would strengthen the Discussion.

6. PLOS authors have the option to publish the peer review history of their article (what does this mean?). If published, this will include your full peer review and any attached files.

Reviewer #1: No

Reviewer #2: No

---

## [Author Response · Author response to Decision Letter 0]

17 Apr 2021

A rebuttal letter, with point-by-point responses to editors and the reviewers' comments is provided with a file name response to reviewers in the attached file.

---

## [Decision Letter · Decision Letter 1]

1 Jun 2021

PONE-D-20-34013R1

Health care providers’ perceptions and experiences related to Midwife-led continuity of care - a qualitative study

PLOS ONE

Dear Dr. Beshah,

Thank you for submitting your manuscript to PLOS ONE. After careful consideration, we feel that it has merit but does not fully meet PLOS ONE’s publication criteria as it currently stands. Therefore, we invite you to submit a revised version of the manuscript that addresses the points raised during the review process.

We look forward to receiving your revised manuscript.

Kind regards,

Bernadette Watson, Ph.D.

Academic Editor

PLOS ONE

Journal Requirements:

Additional Editor Comments (if provided):

As you will see I have received comprehensive comments from one of the Reviewers. I think these suggestoion are comprehensive are substantial and would ask that you address them fully. Bear in mind comments re Q9.

Reviewers' comments:

Reviewer's Responses to Questions

**Comments to the Author**

1. If the authors have adequately addressed your comments raised in a previous round of review and you feel that this manuscript is now acceptable for publication, you may indicate that here to bypass the “Comments to the Author” section, enter your conflict of interest statement in the “Confidential to Editor” section, and submit your "Accept" recommendation.

Reviewer #1: All comments have been addressed

2. Is the manuscript technically sound, and do the data support the conclusions?

Reviewer #1: Partly

3. Has the statistical analysis been performed appropriately and rigorously? 

Reviewer #1: N/A

4. Have the authors made all data underlying the findings in their manuscript fully available?

Reviewer #1: Yes

5. Is the manuscript presented in an intelligible fashion and written in standard English?

Reviewer #1: Yes

6. Review Comments to the Author

Reviewer #1: The manuscript is coming along and reads better this version. A few minor ammendments are suggested to improve further for publication standard:

Line 91 & 92 needs reference

Line 98 when did “traditional” obs/hosp care start

Line 147 – for those who birth at home is it midwifery led?

Line 163 Details about research team: a) what are interviewers experience and training in interviewing for data collection?

b) Relationship with the participants? Are any known to researcher? What relationships existed or were established around this work.

Line 172 – overview broad question asked to the group to context for the reader

Line 185 – Were participants given opportunity to provide feedback/ amend/ approve their transcripts prior data analysis?

Line 186 how many & who involved in coding? – I note this was raised by another reviewer also. Your response is provided in the response to reviewers but has not made it back to the manuscript in the methods section (so it appears it is not attended).

Line 215 – table – include information about how long midwives or if midwives have experience in a midwifery led model of care. This is important as later midwives discuss the success of this and would demonstrate expertise in delivery of this model of care.

Line 217+ Can you demonstrate support for this overarching themes with participant quotes?

Terminology “non-midwife” do you mean doctor? Needs to be consistent thought the whole manuscript.

Q9 General comments not all of these have made it to the manuscript. Some are out of order eg authors responsible for data analysis should be in the methods section – it is included in list of author contributions.

7. PLOS authors have the option to publish the peer review history of their article (what does this mean?). If published, this will include your full peer review and any attached files.

Reviewer #1: No

---

## [Author Response · Author response to Decision Letter 1]

13 Jun 2021

Answer to the editor's comments:

A rebuttal letter, with point-by-point responses to yours and the reviewers' comments is provided in this revised version of the manuscript. The following order has been applied for the submission: Response to Reviewers; Revised Manuscript with Track Changes and Manuscript without track change. 

Journal Requirements:

Answer: yes we revised the reference list based on the journal reference style and we used endnote reference software. We have also tried to replace some of the old references with the recent and most appropriate one. We have checked all the references for retracted paper and all reference lists we used were not retracted. Based on our recent search we have replaced the following reference with the recent reference.

1. Hatem M, Sandall J, Devane D, Soltani H, Gates S. Midwife‐led versus other models of care for childbearing women. Cochrane database of systematic reviews. 2008;4. doi: 10.1002/14651858.CD004667. is replaced by Sandall J, Soltani H, Gates S, Shennan A, Devane D. Midwife-led continuity models versus other models of care for childbearing women. The Cochrane database of systematic reviews. 2013;(8):Cd004667. Epub 2013/08/22. doi: 10.1002/14651858.CD004667.pub3. PubMed PMID: 23963739.

2. World Health Organization, UNICEF. Reduction of maternal mortality: a joint WHO/UNFPA/UNICEF/World Bank Statement. 1999. Is replaced by Muna A, Mohamed A , Zalha A, Sarah BZ, Luc B, Mathieu B, et al. the state of world midwifery report. UNFPA, ICM, WHO, 2021.

3. Reference number 13: PMNCH, WHO, and WB, AHPSR. Success factors for women's and children's health: policy and program highlights from 10 fast-track countries. Geneva: WHO. 2014. Is remuved and Williams K, Lago L, Lainchbury A, Eagar KJM. Mothers’ views of caseload midwifery and the value of continuity of care at an Australian regional hospital. 2010;26(6):615-21. Is added to the reference list 

4. Reference number 45: McCourt C, Page L, Hewison J, A. V. Evaluation of one-to- one midwifery : women responses to care. Birth. 1998 Jun;25(2):73-80. doi: https://doi.org/10.1046/j.1523-536x.1998.00073. Is remuved from the reference list because the idea explained by this article can be adressed by other references used in the manuscript. 

Additional Editor Comments (if provided): 

As you will see I have received comprehensive comments from one of the Reviewers. I think these suggestion are comprehensive are substantial and would ask that you address them fully. Bear in mind comments re Q9.

Answer: Thanks and we made revision on the revised manuscript using track changes based the suggestions provided by the reviewers and also we prepared point by point responses for further elaboration on the questions in the following pages.

Letter to reviewers 

Review Comments followed by answers 

Reviewer #1: The manuscript is coming along and reads better this version. A few minor amendments are suggested to improve further for publication standard:

Answer: Thanks for reading and providing us with helpful insights. All your valuable comments have been taken seriously and made correction on the revised manuscript accordingly.

Line 91 & 92 needs reference

Answer: thank you so much for your valuable insights and correction is made on the manuscript by referencing the original source. These are the references we used in the revised manuscript.

1. Sandall J, Soltani H, Gates S, Shennan A, Devane D. Midwife‐led continuity models versus other models of care for childbearing women. Cochrane database of systematic reviews. 2016;(4). doi: https://doi.org/10.1002/14651858.CD004667.pub5.

2. Zeitlin J, Mohangoo AD, Delnord M, Cuttini M. The second European Perinatal Health Report: documenting changes over 6 years in the health of mothers and babies in Europe. J Epidemiol Community Health. 2013;67(12):983-5. Epub 2013/09/21. doi: 10.1136/jech-2013-203291. PubMed PMID: 24052513.

Line 98 when did “traditional” obs/hosp care start

Answer: In Ethiopian context the only practiced model of care is known as shared model of care (where every health care provider assigned in maternity unit will provide the care for the mother). And this kind of model of care was practiced since the establishment of modern medicine in Ethiopia. 

Line 147 – for those who birth at home is it midwifery led?

Answer: No, for pregnant mother who delivered at home; the traditional birth attendants or relatives will attend her birth. In Ethiopian context Midwives are assigned only at health facilities starting from Health center to referral hospitals. Hence, if the pregnant mother gave birth at home there will not be the chance to be attended by the midwives and other health care providers. This is why we tried to investigate the effectiveness of midwife-led continuity of care model to reduce the rate of home birth. As we have clearly put the percentage of institutional delivery in the study area only 53% of pregnant mother gave birth at the health facility, the remaining 47% of them gave birth at home nonskilled birth attendants. 

Line 163 Details about research team: a) what are interviewers experience and training in interviewing for data collection?

Answer: the principal investigator used to collect the data with two midwifery professionals with MSc in maternal and reproductive health background (one for note taker and the other recorder). But, the principal investigator facilitated the interview with the participants. For this purpose the data collector trained about the interview questions and they practiced it from other similar participants before the actual data collections. Hence, we made correction on the revised manuscript and it is stated in the data collection section of the manuscript about the data collector’s expertise on data collection and that training was given to them before the data collection on the revised version of the manuscript. 

b) Relationship with the participants? Are any known to researcher? What relationships existed or were established around this work.

Answer: thanks for this and we made correction on the revised manuscript as both the principal investigator and the data collectors do not have any relationship with the study participants. All the data collectors are not know to the participants and have no any previous relationship that would affect the data collection process. And of course the data collectors assured the anonymity of the data for the participants before they began the interview. 

Line 172 – overview broad question asked to the group to context for the reader

Answer: thanks for this and we have attached as a supplementary file the interview and FGD guide we used in this study. We used the following questions as an introductory question to provoke discussion: “would you please explain about the type of model of maternal health care practiced in your hospitals and How do you understand the concept of model of maternal health care?” and we continue our interview based on the participants response by using probing questions until we gate reach data on the research question we want to answer.

Line 185 – Were participants given opportunity to provide feedback/ amend/ approve their transcripts prior data analysis?

Answer: yes since the principal investigator undertake the interview there was a chance to transcribe the audio tape recode before we collect the other FGD and interview.

Line 186 how many & who involved in coding? – I note this was raised by another reviewer also. Your response is provided in the response to reviewers but has not made it back to the manuscript in the methods section (so it appears it is not attended).

Answer: thanks for your insight and Code and meaning units were identified by three of the investigators (SH, HL and KC) and we set together and decide on the identified meaning units. And it is included in the revised version of the manuscript.

Line 215 – table – include information about how long midwives or if midwives have experience in a midwifery led model of care. This is important as later midwives discuss the success of this and would demonstrate expertise in delivery of this model of care.

Answer: since there is no well-established and recognized midwife-led continuity of care model in our country we just take the experience of midwives working in maternity unit. But, informally the midwife can give midwife-led continuity of care for example the midwife who provide the antenatal care might have the chance to attend the delivery of the mother and early postnatal care. But this would happen rarely because this would happen when the antenatal care midwife assigns in labour and delivery room during night time or weekend. Due to this reason considering the midwives experience in a midwife-led continuity of care would be difficult. 

Line 217+ Can you demonstrate support for this overarching themes with participant quotes?

Answer: thank you so much for this and the following quote is used and included in the revised manuscript: “As a professional midwife I would be happy if I can give all antenatal, labour and delivery and postnatal care for mothers using the continuum of care model. But, the reality is if I am assigned at antenatal care clinic I would only provide antenatal care; the other unit of care will be covered by other professionals” (midwife code 1 Mehal Meda Hospital).

Terminology “non-midwife” do you mean doctor? Needs to be consistent thought the whole

manuscript.

Answer: With regard to “non-midwife” we used this terminology to represent other health care providers out of midwives who are working in maternal health care unit. It would include medical doctors, integrated emergency surgical officers, etc. and we include this in the revised manuscript at the sample size subsection 

Q9 General comments not all of these have made it to the manuscript. Some are out of order eg authors responsible for data analysis should be in the methods section – it is included in list of author contributions.

Answer: we have included author’s contribution in the method section on the revised manuscript. We tried to include all the comments provided by the reviewers in the revised manuscript. 

Thank you so much once again for your valuable comments and time you spent to evaluate our research work.

With kind regards,

Solomon Hailemeskel (PI)

---

## [Editor Report · Decision Letter 2]

21 Jun 2021

PONE-D-20-34013R2

Health care providers’ perceptions and experiences related to Midwife-led continuity of care - a qualitative study

PLOS ONE

Dear Dr. Beshah,

Thank you for submitting your manuscript to PLOS ONE. After careful consideration, we feel that it has merit but does not fully meet PLOS ONE’s publication criteria as it currently stands. Therefore, we invite you to submit a revised version of the manuscript that addresses the points raised during the review process.

We look forward to receiving your revised manuscript.

Kind regards,

Bernadette Watson, Ph.D.

Academic Editor

PLOS ONE

Journal Requirements:

Additional Editor Comments (if provided):

As you can see one of the reviewers acknowledges the improvement in the paper after your careful edits. However, there are still enough concerns that suggest a second minor revisions is required.

I know this is frustrating but the paper will be a superior product with the suggested revisions.
---

## [Author Response · Author response to Decision Letter 2]

9 Jul 2021

Answer to the editor's comments:

A rebuttal letter, with point-by-point responses to yours and the reviewers' comments is provided in this revised version of the manuscript. The following order has been applied for the submission: Response to Reviewers; Revised Manuscript with Track Changes and Manuscript without track change. 

Journal Requirements:

Answer: yes we revised the reference list based on the journal reference style and we used endnote reference software. We have also tried to replace some of the old references with the recent and most appropriate one. We have checked all the references for retracted paper and all reference lists we used were not retracted. Based on our recent search we have replaced the following reference with the recent reference.

1. Hatem M, Sandall J, Devane D, Soltani H, Gates S. Midwife‐led versus other models of care for childbearing women. Cochrane database of systematic reviews. 2008;4. doi: 10.1002/14651858.CD004667. is replaced by Sandall J, Soltani H, Gates S, Shennan A, Devane D. Midwife-led continuity models versus other models of care for childbearing women. The Cochrane database of systematic reviews. 2013;(8):Cd004667. Epub 2013/08/22. doi: 10.1002/14651858.CD004667.pub3. PubMed PMID: 23963739.

2. World Health Organization, UNICEF. Reduction of maternal mortality: a joint WHO/UNFPA/UNICEF/World Bank Statement. 1999. Is replaced by Muna A, Mohamed A , Zalha A, Sarah BZ, Luc B, Mathieu B, et al. the state of world midwifery report. UNFPA, ICM, WHO, 2021.

3. Reference number 13: PMNCH, WHO, and WB, AHPSR. Success factors for women's and children's health: policy and program highlights from 10 fast-track countries. Geneva: WHO. 2014. Is remuved and Williams K, Lago L, Lainchbury A, Eagar KJM. Mothers’ views of caseload midwifery and the value of continuity of care at an Australian regional hospital. 2010;26(6):615-21. Is added to the reference list 

4. Reference number 45: McCourt C, Page L, Hewison J, A. V. Evaluation of one-to- one midwifery : women responses to care. Birth. 1998 Jun;25(2):73-80. doi: https://doi.org/10.1046/j.1523-536x.1998.00073. Is removed from the reference list because the idea explained by this article can be addressed by other references used in the manuscript. 

Additional Editor Comments (if provided): 

As you will see I have received comprehensive comments from one of the Reviewers. I think these suggestion are comprehensive are substantial and would ask that you address them fully. Bear in mind comments re Q9.

Answer: Thanks and we made revision on the revised manuscript using track changes based the suggestions provided by the reviewers and also we prepared point by point responses for further elaboration on the questions in the following pages.

Letter to reviewers 

Review Comments followed by answers 

Reviewer #1: The manuscript is coming along and reads better this version. A few minor amendments are suggested to improve further for publication standard:

Answer: Thanks for reading and providing us with helpful insights. All your valuable comments have been taken seriously and made correction on the revised manuscript accordingly.

Line 91 & 92 needs reference

Answer: thank you so much for your valuable insights and correction is made on the manuscript by referencing the original source. These are the references we used in the revised manuscript.

1. Sandall J, Soltani H, Gates S, Shennan A, Devane D. Midwife‐led continuity models versus other models of care for childbearing women. Cochrane database of systematic reviews. 2016;(4). doi: https://doi.org/10.1002/14651858.CD004667.pub5.

2. Zeitlin J, Mohangoo AD, Delnord M, Cuttini M. The second European Perinatal Health Report: documenting changes over 6 years in the health of mothers and babies in Europe. J Epidemiol Community Health. 2013;67(12):983-5. Epub 2013/09/21. doi: 10.1136/jech-2013-203291. PubMed PMID: 24052513.

Line 98 when did “traditional” obs/hosp care start

Answer: In Ethiopian context the only practiced model of care is known as shared model of care (where every health care provider assigned in maternity unit will provide the care for the mother). And this kind of model of care was practiced since the establishment of modern medicine in Ethiopia. 

Line 147 – for those who birth at home is it midwifery led?

Answer: No, for pregnant mother who delivered at home; the traditional birth attendants or relatives will attend her birth. In Ethiopian context Midwives are assigned only at health facilities starting from Health center to referral hospitals. Hence, if the pregnant mother gave birth at home there will not be the chance to be attended by the midwives and other health care providers. This is why we tried to investigate the effectiveness of midwife-led continuity of care model to reduce the rate of home birth. As we have clearly put the percentage of institutional delivery in the study area only 53% of pregnant mother gave birth at the health facility, the remaining 47% of them gave birth at home nonskilled birth attendants. 

Line 163 Details about research team: a) what are interviewers experience and training in interviewing for data collection?

Answer: the principal investigator used to collect the data with two midwifery professionals with MSc in maternal and reproductive health background (one for note taker and the other recorder). But, the principal investigator facilitated the interview with the participants. For this purpose the data collector trained about the interview questions and they practiced it from other similar participants before the actual data collections. Hence, we made correction on the revised manuscript and it is stated in the data collection section of the manuscript about the data collector’s expertise on data collection and that training was given to them before the data collection on the revised version of the manuscript. 

b) Relationship with the participants? Are any known to researcher? What relationships existed or were established around this work.

Answer: thanks for this and we made correction on the revised manuscript as both the principal investigator and the data collectors do not have any relationship with the study participants. All the data collectors are not know to the participants and have no any previous relationship that would affect the data collection process. And of course the data collectors assured the anonymity of the data for the participants before they began the interview. 

Line 172 – overview broad question asked to the group to context for the reader

Answer: thanks for this and we have attached as a supplementary file the interview and FGD guide we used in this study. We used the following questions as an introductory question to provoke discussion: “would you please explain about the type of model of maternal health care practiced in your hospitals and How do you understand the concept of model of maternal health care?” and we continue our interview based on the participants response by using probing questions until we gate reach data on the research question we want to answer.

Line 185 – Were participants given opportunity to provide feedback/ amend/ approve their transcripts prior data analysis?

Answer: yes since the principal investigator undertake the interview there was a chance to transcribe the audio tape recode before we collect the other FGD and interview.

Line 186 how many & who involved in coding? – I note this was raised by another reviewer also. Your response is provided in the response to reviewers but has not made it back to the manuscript in the methods section (so it appears it is not attended).

Answer: thanks for your insight and Code and meaning units were identified by three of the investigators (SH, HL and KC) and we set together and decide on the identified meaning units. And it is included in the revised version of the manuscript.

Line 215 – table – include information about how long midwives or if midwives have experience in a midwifery led model of care. This is important as later midwives discuss the success of this and would demonstrate expertise in delivery of this model of care.

Answer: since there is no well-established and recognized midwife-led continuity of care model in our country we just take the experience of midwives working in maternity unit. But, informally the midwife can give midwife-led continuity of care for example the midwife who provide the antenatal care might have the chance to attend the delivery of the mother and early postnatal care. But this would happen rarely because this would happen when the antenatal care midwife assigns in labour and delivery room during night time or weekend. Due to this reason considering the midwives experience in a midwife-led continuity of care would be difficult. 

Line 217+ Can you demonstrate support for this overarching themes with participant quotes?

Answer: thank you so much for this and the following quote is used and included in the revised manuscript: “As a professional midwife I would be happy if I can give all antenatal, labour and delivery and postnatal care for mothers using the continuum of care model. But, the reality is if I am assigned at antenatal care clinic I would only provide antenatal care; the other unit of care will be covered by other professionals” (midwife code 1 Mehal Meda Hospital).

Terminology “non-midwife” do you mean doctor? Needs to be consistent thought the whole

manuscript.

Answer: With regard to “non-midwife” we used this terminology to represent other health care providers out of midwives who are working in maternal health care unit. It would include medical doctors, integrated emergency surgical officers, etc. and we include this in the revised manuscript at the sample size subsection 

Q9 General comments not all of these have made it to the manuscript. Some are out of order eg authors responsible for data analysis should be in the methods section – it is included in list of author contributions.

Answer: we have included author’s contribution in the method section on the revised manuscript. We tried to include all the comments provided by the reviewers in the revised manuscript. 

Thank you so much once again for your valuable comments and time you spent to evaluate our research work.

With kind regards,

Solomon Hailemeskel (PI)

---

## [Decision Letter · Decision Letter 3]

23 Aug 2021

PONE-D-20-34013R3

Health care providers’ perceptions and experiences related to Midwife-led continuity of care - a qualitative study

PLOS ONE

Dear Dr. Beshah,

Thank you for submitting your manuscript to PLOS ONE. After careful consideration, we feel that it has merit but does not fully meet PLOS ONE’s publication criteria as it currently stands. Therefore, we invite you to submit a revised version of the manuscript that addresses the points raised during the review process.

We look forward to receiving your revised manuscript.

Kind regards,

Bernadette Watson, Ph.D.

Academic Editor

PLOS ONE

Journal Requirements:

Additional Editor Comments (if provided):

This paper is very much improved given your attention to the reviewers' earlier comments. Please give serious attention to the comments they raise and return the paper fully revised. I look forward to receiving the revisions.

Reviewers' comments:

Reviewer's Responses to Questions

**Comments to the Author**

1. If the authors have adequately addressed your comments raised in a previous round of review and you feel that this manuscript is now acceptable for publication, you may indicate that here to bypass the “Comments to the Author” section, enter your conflict of interest statement in the “Confidential to Editor” section, and submit your "Accept" recommendation.

Reviewer #1: (No Response)

2. Is the manuscript technically sound, and do the data support the conclusions?

Reviewer #1: Yes

3. Has the statistical analysis been performed appropriately and rigorously? 

Reviewer #1: N/A

4. Have the authors made all data underlying the findings in their manuscript fully available?

Reviewer #1: Yes

5. Is the manuscript presented in an intelligible fashion and written in standard English?

Reviewer #1: No

6. Review Comments to the Author

Reviewer #1: The manuscript continues to be refined and is much improved from earlier verions, so well done. However,

there are still minor ammendments which will improve the quality of your work and experience of the reader.

As per earlier feedback, language needs to be consistent, this occurs in many instances in your manuscript (as follows) so it needs to be accurate if they are midwives and doctors then continue with this terminology. If you include a tracked changed document these will be easily noted.

LINE 30 “25 midwives and 8 emergency surgical officers and medical doctors”

versus

LINE 41 “25 midwives and the group of 8 non-midwives”

Versus

LINE 160 “25 midwives and 8 non-midwives (health care providers other than midwives working in maternal health care unit)”

versus

LINE 171 “ (non-midwife health care providers working in the maternal health care unit”

versus

LINE 219 “25 midwives and 8 medical directors and integrated emergency surgical officers”

versus

table 1 : “Medical Doctor Integrated Emergency Surgical Officer (IESO)”

LINE 299 Non-midwife

LINE 325 Non-midwife

LINE 3xx Non-midwife

LINE 352 Non-midwife

LINE 357 non-midwife health care providers

LINE 369 non-midwife health care provider.

LINE 374 non-midwife

LINE 405 non-midwife = medical model

Conclusion:

LINE 30 seems most accurate

LINE 87 do you have a reference for your improved care outcomes?

LINE 124 So this study explores midwives and medical doctors perceptions. Be clear this is who is included…. Unless there are other health care professionals included? Then specify these.

LINE 162. Needs re-writing for clarity “At this saturation point, the study participants were unable to provide any additional new ideas”.

LINE 179 edit for clarity

LINE 188 Not clear if returned to participants for checking?

LINE 223 Comment if this is the usual distribution of genders for midwives (ie are they mostly male???)

LINE 493 “This study is the first to describe the experience and perceptions of” use which ever term you choose consistently and conclude with it here eg Midwives and doctors

7. PLOS authors have the option to publish the peer review history of their article (what does this mean?). If published, this will include your full peer review and any attached files.

Reviewer #1: No

---

## [Author Response · Author response to Decision Letter 3]

3 Sep 2021

Answer to the editor's comments:

A rebuttal letter, with point-by-point responses to yours and the reviewers' comments is provided in this revised version of the manuscript. The following order has been applied for the submission: Response to Reviewers; Revised Manuscript with Track Changes and Manuscript without track change. 

Journal Requirements:

Answer: yes we revised the reference list based on the journal reference style and we used endnote reference software. We have also tried to replace some of the old references with the recent and most appropriate one. We have checked all the references for retracted paper and all reference lists we used were not retracted. And we have actually corrected the reference lists based on pervious comments and we now check it again for any. 

Letter to reviewers 

Review Comments followed by answers 

Reviewer #1: The manuscript continues to be refined and is much improved from earlier versions, so well done. However, there are still minor amendments which will improve the quality of your work and experience of the reader.

As per earlier feedback, language needs to be consistent, this occurs in many instances in your manuscript (as follows) so it needs to be accurate if they are midwives and doctors then continue with this terminology. If you include a tracked changed document these will be easily noted.

Author’s response: thank you so much for your valuable comments. For your general understanding we interview all health care providers working in maternal health care unit. In Ethiopian context the health care providers working in maternal health care unit are midwives, nurses, medical doctors and emergency surgical officers. For this reason we tried to classify the health care providers as midwives (who are professional midwives working in the maternal health care unit) and non-midwives (health care providers other than midwives who are currently working in maternal health care unit. This includes: medical doctors and emergency surgical officers). As you remember our research question was to answer the perception and experience of health care providers working in maternal health care unit about midwife-led continuity of care model. During our visit at the health facility we found 12-15 midwives in each hospital and 2 emergency surgical officers and 2 medical doctors working in maternal health care unit. As a result of this we conducted FGD for midwives and individual in-depth interview for emergency surgical officers and medical doctors. Finally, for write up and communication purpose we classify them as midwife and non-midwife. The main reason we use the term non-midwife was the health care providers out of midwives are not purely a medical doctors (some of them are medical doctors and some of them are emergency surgical officers). Hence, if this classification creates confusion for readers we better to keep the classification as midwife and integrated emergency surgical officers and medical doctors as it is. 

LINE 30 “25 midwives and 8 emergency surgical officers and medical doctors”

Author’s response: Thanks for this and we keep is as it is 

Versus

LINE 41 “25 midwives and the group of 8 non-midwives”

Author’s response: Thanks for this and we replaced the word non-midwives by emergency surgical officers and medical doctors 

Versus

LINE 160 “25 midwives and 8 non-midwives (health care providers other than midwives working in maternal health care unit)”

Author’s response: Thanks for this and we replaced the word non-midwives by emergency surgical officers and medical doctors

versus

LINE 171 “ (non-midwife health care providers working in the maternal health care unit”

Author’s response: Thanks for this and we replaced the word non-midwives by emergency surgical officers and medical doctors

Versus

LINE 219 “25 midwives and 8 medical directors and integrated emergency surgical officers”

Author’s response: Thanks for this and we keep as it is 

Versus

table 1 : “Medical Doctor Integrated Emergency Surgical Officer (IESO)”

Author’s response: Thanks for this and we keep as it is 

LINE 299 Non-midwife

Author’s response: Thanks for this and we replaced the word non-midwives by emergency surgical officers 

LINE 325 Non-midwife

Author’s response: Thanks for this and we replaced the word non-midwives by medical doctors 

LINE 3xx Non-midwife

Author’s response: Thanks for this and we replaced the word non-midwives by emergency surgical officers

LINE 352 Non-midwife

Author’s response: Thanks for this and we replaced the word non-midwives by emergency surgical officers and medical doctors 

LINE 357 non-midwife health care providers

Author’s response: Thanks for this and we replaced the word non-midwives by emergency surgical officers and medical doctors

LINE 369 non-midwife health care provider.

Author’s response: Thanks for this and we replaced the word non-midwives by emergency surgical officers and medical doctors

LINE 374 non-midwife

Author’s response: Thanks for this and we replaced the word non-midwives by medical doctors 

LINE 405 non-midwife = medical model

Author’s response: thanks for the comments and we replaced the word non-midwife-led model by existed model of care (shared model of care). 

Conclusion:

LINE 30 seems most accurate

LINE 87 do you have a reference for your improved care outcomes?

Author’s response: thanks for the comment and appropriate reference is inserted in the document 

LINE 124 So this study explores midwives and medical doctors perceptions. Be clear this is who is included…. Unless there are other health care professionals included? Then specify these.

Author’s response: thanks for your comments. Just to be clear we tried to include all health care providers who are working in maternal health care unit. For this reason we interviewed medical doctors and integrated emergency surgical officers. And we do FGD with midwives. The main reason we cannot categorize the participants as midwives and medical doctors were the integrated emergency surgical officers are not a medical doctor. They specialize from other health science professional like nurses and health officers at master’s level to do emergency surgical procedures like emergency caesarian section and acute abdomen. Due to this reason we choose the word non-midwife for them. However we replaced the pervious word by integrated emergency surgical officers and medical doctors. And correction made accordingly 

LINE 162. Needs re-writing for clarity “At this saturation point, the study participants were unable to provide any additional new ideas”.

Author’s response: thanks for the comment and it is rewrite like this “The data collection was completed when no new/unique information was emerged”

LINE 179 edit for clarity

Author’s response: The discussion was initiated by asking the study participants to explain about the type of model of maternal health care practiced in their hospitals and how do they understand model of maternal health care? This sentence was the first open ended question asked for study participants to initiate the discussion. Then following their response we ask them a probing question to explore more. 

LINE 188 Not clear if returned to participants for checking?

Author’s response: All the interviews were performed and coded initially by the first author, who was fluent speaker in original language (Amharic). Hence, we did not return the translated word to the participants. 

LINE 223 Comment if this is the usual distribution of genders for midwives (ie are they mostly male???)

Author’s response: thank you so much for this and it seems stranger. But, according to state of Ethiopian midwives study the distribution of female midwives was 67.3%. However, we found small number of female midwives in our study area during the time of interview. 

LINE 493 “This study is the first to describe the experience and perceptions of” use which ever term you choose consistently and conclude with it here eg Midwives and doctors

Author’s response: thanks for the comment. We used the term Ethiopian health care providers because we include all types of health care providers working in maternal health care unit. But to be more clear on the conclusion we change the word and replaced by “Midwives, Integrated emergency surgical officers and medical doctors”

Thank you so much once again for your valuable comments and time you spent to evaluate our research work.

With kind regards,

Solomon Hailemeskel (PI)

---

## [Editor Report · Decision Letter 4]

23 Sep 2021

Health care providers’ perceptions and experiences related to Midwife-led continuity of care - a qualitative study

PONE-D-20-34013R4

Dear Dr. Beshah,

We’re pleased to inform you that your manuscript has been judged scientifically suitable for publication and will be formally accepted for publication once it meets all outstanding technical requirements.

Kind regards,

Bernadette Watson, Ph.D.

Academic Editor

PLOS ONE

Additional Editor Comments (optional):

Thank you for your attention to these final issues. The paper reads very well.
---

## [Editor Report · Acceptance letter]

7 Oct 2021

PONE-D-20-34013R4 

Health care providers’ perceptions and experiences related to Midwife-led continuity of care - a qualitative study 

Dear Dr. Hailemeskel:

I'm pleased to inform you that your manuscript has been deemed suitable for publication in PLOS ONE. Congratulations! Your manuscript is now with our production department. 

Kind regards, 

on behalf of

Dr. Bernadette Watson 

Academic Editor

PLOS ONE